# Semantical and geometrical protein encoding toward enhanced bioactivity and thermostability

Yang Tan[1,2,3,4†], Bingxin Zhou[1,3,5,6*†], Lirong Zheng[5], Guisheng Fan[2], Liang Hong[1,3,4,5,6*]

[1]Shanghai-Chongqing Institute of Artificial Intelligence, Shanghai Jiao Tong University, Chongqing, China; [2]School of Information Science and Engineering, East China University of Science and Technology, Shanghai, China; [3]Zhangjiang Institute for Advanced Study, Shanghai Jiao Tong University, Shanghai, China; [4]Shanghai Artificial Intelligence Laboratory, Shanghai, China; [5]Shanghai Jiao Tong University, Institute of Natural Sciences, Shanghai, China; [6]Shanghai National Center for Applied Mathematics (SJTU Center), Shanghai Jiao Tong University, Shanghai, China

*For correspondence:
bingxin.zhou@sjtu.edu.cn (BZ);
hongl3liang@sjtu.edu.cn (LH)

†These authors contributed equally to this work

Competing interest: The authors declare that no competing interests exist.

## eLife Assessment

ProtSSN is a **valuable** approach that generates protein embeddings by integrating sequence and structural information, demonstrating improved prediction of mutation effects on thermostability compared to competing models. The evidence supporting the authors' claims is **compelling**, with well-executed comparisons. This work will be of particular interest to researchers in bioinformatics and structural biology, especially those focused on protein function and stability.

**Abstract** Protein engineering is a pivotal aspect of synthetic biology, involving the modification of amino acids within existing protein sequences to achieve novel or enhanced functionalities and physical properties. Accurate prediction of protein variant effects requires a thorough understanding of protein sequence, structure, and function. Deep learning methods have demonstrated remarkable performance in guiding protein modification for improved functionality. However, existing approaches predominantly rely on protein sequences, which face challenges in efficiently encoding the geometric aspects of amino acids' local environment and often fall short in capturing crucial details related to protein folding stability, internal molecular interactions, and bio-functions. Furthermore, there lacks a fundamental evaluation for developed methods in predicting protein thermostability, although it is a key physical property that is frequently investigated in practice. To address these challenges, this article introduces a novel pre-training framework that integrates sequential and geometric encoders for protein primary and tertiary structures. This framework guides mutation directions toward desired traits by simulating natural selection on wild-type proteins and evaluates variant effects based on their fitness to perform specific functions. We assess the proposed approach using three benchmarks comprising over 300 deep mutational scanning assays. The prediction results showcase exceptional performance across extensive experiments compared to other zero-shot learning methods, all while maintaining a minimal cost in terms of trainable parameters. This study not only proposes an effective framework for more accurate and comprehensive predictions to facilitate efficient protein engineering, but also enhances the in silico assessment system for future deep learning models to better align with empirical requirements. The PyTorch implementation is available at https://github.com/ai4protein/ProtSSN.

## Introduction

The diverse roles proteins play in nature necessitate their involvement in varied bio-functions, such as catalysis, binding, scaffolding, and transport. Advances in science and technology have positioned proteins, particularly those with desired catalytic and binding properties, to pivotal positions in medicine, biology, and scientific research (*Zhou et al., 2024a*). While wild-type proteins exhibit optimal bio-functionality in their native environments, industrial applications often demand adaptation to conditions such as high temperature, high pressure, strong acidity, and strong alkalinity. The constrained efficacy of proteins in meeting the stringent requirements of industrial functioning environments hinders their widespread applications (*Woodley, 2013*; *Ismail et al., 2021*; *Jiang et al., 2023*). Consequently, there arises a need to engineer these natural proteins to enhance their functionalities, aligning them with the demands of both industrial and scientific applications.

Analyzing the relationship between protein sequence and function yields valuable insights for engineering proteins with new or enhanced functions in synthetic biology. The intrinsic goal of protein engineering is to unveil highly functional sequences by navigating the intricate, high-dimensional surface of the fitness landscape (*Yang et al., 2019*), which delineates the relationship between amino acid (AA) sequences and the desired functional state. Predicting the effects of AA substitutions, insertions, and deletions (*Riesselman et al., 2018*; *Gray et al., 2018*; *Notin et al., 2022a*) in proteins, that is, mutation, thus allows researchers to dissect how changes in the AA sequence can impact the protein's catalytic efficiency, stability, and binding affinity (*Shin et al., 2021*). However, the extensive, non-contiguous search space of AA combinations poses challenges for conventional methods, such as rational design (*Aprile et al., 2020*) or directed evolution (*Wang et al., 2021*), in efficiently and effectively identifying protein variants with desired properties. In response, deep learning has emerged as a promising solution that proposes favorable protein variants with high fitness.

Deep learning approaches have been instrumental in advancing scientific insights into proteins, predominantly categorized into sequence-based and structure-based methods. Autoregressive protein language models (*Notin et al., 2022a*; *Madani et al., 2023*) interpret AA sequences as raw text to generate with self-attention mechanisms (*Vaswani et al., 2017*). Alternatively, masked language modeling objectives develop attention patterns that correspond to the residue–residue contact map of the protein (*Meier et al., 2021*; *Rao et al., 2021b*; *Rives et al., 2021*; *Vig et al., 2021*; *Brandes et al., 2022*; *Lin et al., 2023*). Other methods start from a multiple sequence alignment, summarizing the evolutionary patterns in target proteins (*Riesselman et al., 2018*; *Frazer et al., 2021*; *Rao et al., 2021a*). These methods result in a strong capacity for discovering the hidden protein space. However, the many-to-one relationship of both sequence-to-structure and structure-to-function projections requires excessive training input or substantial learning resources, which raises concerns regarding the efficiency of these pathways when navigating the complexities of the vast sequence space associated with a target function. Moreover, the overlooked local environment of proteins hinders the model's ability to capture structure-sensitive properties that impact protein's thermostability, interaction with substrates, and catalytic process (*Zhao et al., 2022*; *Koehler Leman et al., 2023*). Alternatively, structure-based modeling methods serve as an effective enhancement to complement sequence-oriented inferences for proteins with their local environment (*Tan et al., 2024e*; *Zhou et al., 2024c*; *Yi et al., 2024*; *Zhou et al., 2024b*; *Tan et al., 2024c*). Given that core mutations often induce functional defects through subtle disruptions to structure or dynamics (*Roscoe et al., 2013*), incorporating protein geometry into the learning process can offer valuable insights into stabilizing protein functioning. Recent efforts have been made to encode geometric information of proteins for topology-sensitive tasks such as molecule binding (*Jin et al., 2021*; *Myung et al., 2022*; *Kong et al., 2023*) and protein properties prediction (*Zhang et al., 2022*; *Tan et al., 2024d*; *Tan et al., 2024a*). Nevertheless, structure encoders fall short in capturing non-local connections for AAs beyond their contact region and overlook correlations that do not conform to the 'structure–function determination' heuristic.

There is a pressing need to develop a novel framework that overcomes the limitations inherent in individual implementations of sequence or structure-based investigations. To this end, we introduce ProtSSN to assimilate the semantics and topology of **Prot**eins from their **S**equence and **S**tructure with deep neural **N**etworks. ProtSSN incorporates the intermediate state of protein structures and facilitates the discovery of an efficient and effective trajectory for mapping protein sequences to functionalities. The developed model extends the generalization and robustness of self-supervised protein

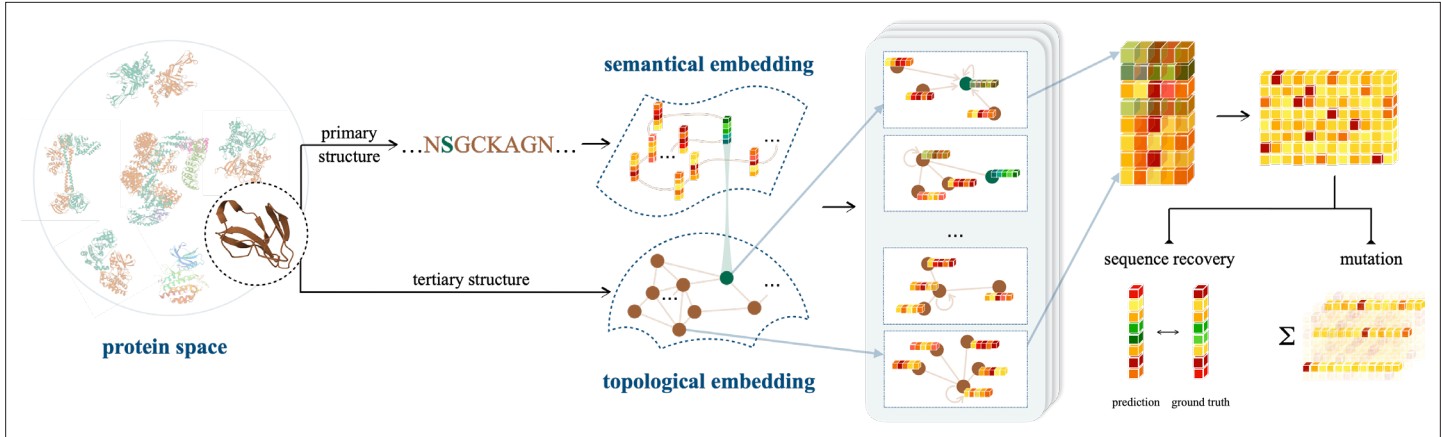

**Figure 1.** An illustration of ProtSSN that extracts the semantical and geometrical characteristics of a protein from its sequentially ordered global construction and spatially gathered local contacts with protein language models and equivariant graph neural networks. The encoded hidden representation can be used for downstream tasks such as variants effect prediction that recognizes the impact of mutating a few sites of a protein on its functionality.

language models while maintaining low computational costs, thereby facilitating self-supervised training and task-specific customization. A funnel-shaped learning pipeline, as depicted in *Figure 1*, is designed due to the limited availability of protein crystal structures compared to observed protein sequences. Initially, the linguistic embedding establishes the semantic and grammatical rules in AA chains by inspecting millions of protein sequences. The topological embedding then enhances the interactions among locally connected AAs. Since a geometry can be placed or observed by different angles and positions in space, we represent proteins' topology by graphs and enhance the model's robustness and efficiency with a rotation and translation equivariant graph representation learning scheme.

The pre-trained ProtSSN demonstrates its feasibility across a broad range of mutation effect prediction benchmarks covering catalysis, interaction, and thermostability. Mutation effects prediction serves as a common in silico assessment for evaluating the capability of an established deep learning framework in identifying favorable protein variants. With the continuous evolution of deep mutation scanning techniques and other high-throughput technologies, benchmark datasets have been curated to document fitness changes in mutants across diverse proteins with varying degrees of combinatorial and random modifications (*Moal and Fernández-Recio, 2012*; *Nikam et al., 2021*; *Velecký et al., 2022*). **ProteinGym v1** (*Notin et al., 2023*) comprehends fitness scoring of mutants to catalytic activity and binding affinity across over 200 well-studied proteins of varying deep mutational scanning (DMS) assays and taxa. Each protein includes tens of thousands of mutants documented from high-throughput screening techniques. While **ProteinGym v1** initiates the most comprehensive benchmark dataset for mutants toward different properties enhancement, these fitness scores are normalized and simplified into several classes without further specifications. For instance, a good portion of protein assays is scored by their stability, which, in practice, is further categorized into thermostability, photostability, pH-stability, etc. Moreover, these stability properties should be discussed under certain environmental conditions, such as pH and temperature. Unfortunately, these detailed attributes are excluded in **ProteinGym v1**. Since a trade-off relationship emerges between activity and thermostability (*Zheng et al., 2022*), protein engineering in practice extends beyond enhancing catalytic activities to maintaining or increasing thermostability for prolonged functional lifetimes and applications under elevated temperatures and chemical conditions (*Liu et al., 2019*). Consequently, there is a need to enrich mutation effect prediction benchmarks that evaluate the model's efficacy in capturing variant effects concerning thermostability under distinct experimental conditions. In this study, we address the mutation effect prediction tasks on thermostability with **DTm** and **DDG**, two new single-site mutation benchmarks that measure thermostability using ΔTm and ΔΔG values, respectively. Both benchmarks group experimental assays based on the protein–condition combinations. These two datasets supplement the publicly available DMS assay benchmarks and facilitate ready-to-use assessment for future deep learning methods toward protein thermostability enhancement.

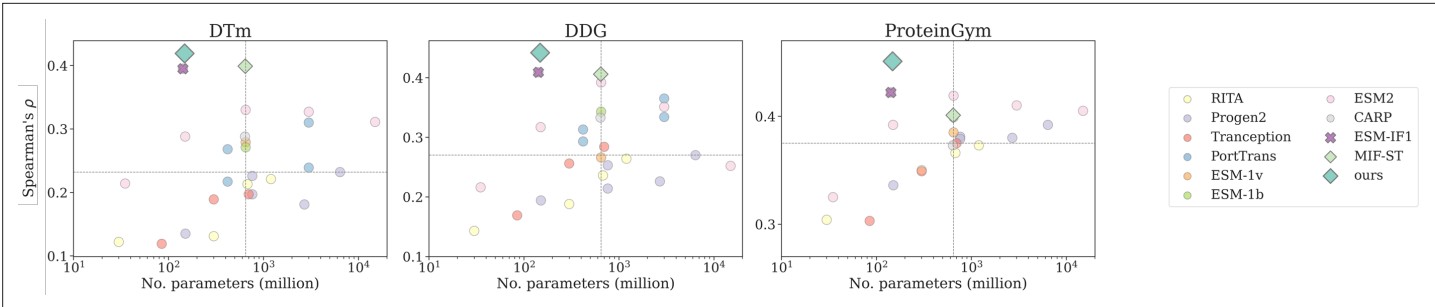

**Figure 2.** Number of trainable parameters and Spearman's $\rho$ correlation on **DTm**, **DDG**, and **ProteinGym v1**, with the medium value located by the dashed lines. Dot, cross, and diamond markers represent sequence-based, structure-based, and sequence-structure models, respectively.

This work fulfills practical necessities in the development of deep learning methods for protein engineering from three distinct perspectives. Firstly, the developed ProtSSN employs self-supervised learning during training to obviate the necessity for additional supervision in downstream tasks. This zero-shot scenario is desirable due to the scarcity of experimental results, as well as the 'cold-start' situation common in many wet lab experiments. Secondly, the trained model furnishes robust and meaningful approximations to the joint distribution of the entire AA chain, thereby augmenting the *epistatic effect* (*Sarkisyan et al., 2016*; *Khersonsky et al., 2018*) in deep mutations. This augmentation stems from the model's consideration of the nonlinear combinatorial effects of AA sites. Additionally, we complement the existing dataset of DMS assay with additional benchmark assays related to protein–environment interactions, specifically focusing on thermostability—an integral objective in numerous protein engineering projects.

## Results

### Variant effect prediction

We commence our investigation by assessing the predictive performance of ProtSSN on four benchmark datasets using state-of-the-art (SOTA) protein learning models. We deploy different versions of ProtSSN that learns $k$NN graphs with $k \in \{10, 20, 30\}$ and $h \in \{512, 768, 1,280\}$ hidden neurons in each of the six *Equivariant Graph Neural Networks* EGNN layers. For all baselines, their performance on **DTm** and **DDG** is reproduced with the official implementation listed in Table 5, and the scores on **ProteinGym v1** are retrieved from https://proteingym.org/benchmarks by *Notin et al., 2023*. As visualized in *Figure 2* and reported in *Table 1*, ProtSSN demonstrated exceptional predictive performance with a significantly smaller number of trainable parameters when predicting the function and thermostability of mutants.

Compared to protein language models (*Figure 2*, colored circles), ProtSSN benefits from abundant structure information to more accurately capture the overall protein characteristics, resulting in enhanced thermostability and thus achieving higher correlation scores on **DTm** and **DDG**. This is attributed to the compactness of the overall architecture as well as the presence of stable local structures such as alpha helices and beta sheets, both of which are crucial to protein thermostability by providing a framework that resists unfolding at elevated temperatures (*Robinson-Rechavi et al., 2006*). Consequently, the other two structures involved models, that is, ESM-if1 and MIF-ST, also exhibit higher performance on the two thermostability benchmarks.

On the other hand, although protein language models can typically increase their scale, that is, the number of trainable parameters, to capture more information from sequences, they cannot fully replace the role of the structure-based analysis. This observation aligns with the mechanism of protein functionality, where local structures (such as binding pockets and catalytic pockets) and the overall structure (such as conformational changes and allosteric effect) are both crucial for binding and catalysis (*Sheng et al., 2014*). Unlike the thermostability benchmarks, the discrepancies between structure-involved models and protein language models are mitigated in **ProteinGym v1** by increasing the model scale. In summary, structure-based and sequence-based methods are suitable for different types of assays, but a comprehensive framework (such as ProtSSN) demonstrates superior overall performance compared to large-scale language models. Moreover, ProtSSN demonstrates

**Table 1.** Spearman's $\rho$ correlation of variant effect prediction by with zero-shot methods on **DTm**, **DDG**, and **ProteinGym v1**.

| Model | Version | # Params (million) | DTm | DDG | ProteinGym v1 Activity | Binding | Expression | Organismal | Stability | Overall fitness |
|---|---|---|---|---|---|---|---|---|---|---|
| Sequence encoder | | | | | | | | | | |
| | Small | 30 | 0.122 | 0.143 | 0.294 | 0.275 | 0.337 | 0.327 | 0.289 | 0.304 |
| | Medium | 300 | 0.131 | 0.188 | 0.352 | 0.274 | 0.406 | 0.371 | 0.348 | 0.350 |
| | Large | 680 | 0.213 | 0.236 | 0.359 | 0.291 | 0.422 | 0.374 | 0.383 | 0.366 |
| RITA | xlarge | 1,200 | 0.221 | 0.264 | 0.402 | 0.302 | 0.423 | 0.387 | 0.445 | 0.373 |
| | Small | 151 | 0.135 | 0.194 | 0.333 | 0.275 | 0.384 | 0.337 | 0.349 | 0.336 |
| | Medium | 764 | 0.226 | 0.214 | 0.393 | 0.296 | 0.436 | 0.381 | 0.396 | 0.380 |
| | Base | 764 | 0.197 | 0.253 | 0.396 | 0.294 | 0.444 | 0.379 | 0.383 | 0.379 |
| | Large | 2700 | 0.181 | 0.226 | 0.406 | 0.294 | 0.429 | 0.379 | 0.396 | 0.381 |
| ProGen2 | xlarge | 6400 | 0.232 | 0.270 | 0.402 | 0.302 | 0.423 | 0.387 | 0.445 | 0.392 |
| | bert | 420 | 0.268 | 0.313 | - | - | - | - | - | - |
| | bert_bfd | 420 | 0.217 | 0.293 | - | - | - | - | - | - |
| | t5_xl_uniref50 | 3000 | 0.310 | 0.365 | - | - | - | - | - | - |
| ProtTrans | t5_xl_bfd | 3000 | 0.239 | 0.334 | - | - | - | - | - | - |
| | Small | 85 | 0.119 | 0.169 | 0.287 | 0.349 | 0.319 | 0.270 | 0.258 | 0.288 |
| | Medium | 300 | 0.189 | 0.256 | 0.349 | 0.285 | 0.409 | 0.362 | 0.342 | 0.349 |
| Tranception | Large | 700 | 0.197 | 0.284 | 0.401 | 0.289 | 0.415 | 0.389 | 0.381 | 0.375 |
| ESM-1v | - | 650 | 0.279 | 0.266 | 0.390 | 0.268 | 0.431 | 0.362 | 0.476 | 0.385 |
| ESM-1b | - | 650 | 0.271 | 0.343 | 0.428 | 0.289 | 0.427 | 0.351 | 0.500 | 0.399 |
| | t12 | 35 | 0.214 | 0.216 | 0.314 | 0.292 | 0.364 | 0.218 | 0.439 | 0.325 |
| | t30 | 150 | 0.288 | 0.317 | 0.391 | 0.328 | 0.425 | 0.305 | 0.510 | 0.392 |
| | t33 | 650 | 0.330 | 0.392 | 0.425 | 0.339 | 0.415 | 0.338 | 0.523 | 0.419 |
| | t36 | 3000 | 0.327 | 0.351 | 0.417 | 0.322 | 0.425 | 0.379 | 0.509 | 0.410 |
| ESM2 | t48 | 15,000 | 0.311 | 0.252 | 0.405 | 0.318 | 0.425 | 0.388 | 0.488 | 0.405 |
| CARP | - | 640 | 0.288 | 0.333 | 0.395 | 0.274 | 0.419 | 0.364 | 0.414 | 0.373 |
| Structure encoder | | | | | | | | | | |
| ESM-if1 | - | 142 | 0.395 | 0.409 | 0.368 | 0.392 | 0.403 | 0.324 | 0.624 | 0.422 |
| Sequence + structure encoder | | | | | | | | | | |
| MIF-ST | - | 643 | 0.400 | 0.406 | 0.390 | 0.323 | 0.432 | 0.373 | 0.486 | 0.401 |
| | Masked | 650 | 0.382 | - | 0.459 | 0.382 | 0.485 | 0.371 | 0.583 | 0.456 |
| SaProt | Unmasked | 650 | 0.376 | 0.359 | 0.450 | 0.376 | 0.460 | 0.372 | 0.577 | 0.447 |
| | k20_h512 | 148 | 0.419 | 0.442 | 0.458 | 0.371 | 0.436 | 0.387 | 0.566 | 0.444 |
| ProtSSN (ours) | Ensemble | 1467 | 0.425 | 0.440 | 0.466 | 0.371 | 0.451 | 0.398 | 0.568 | 0.451 |

The top three are highlighted by first, second, and third.

consistency in providing high-quality fitness predictions of thermostability. We randomly bootstrap 50% of the samples from **DTm** and **DDG** for 10 independent runs, the results are reported In Table 3 for both the average performance and the standard deviation. ProtSSN achieves top performance with minimal variance.

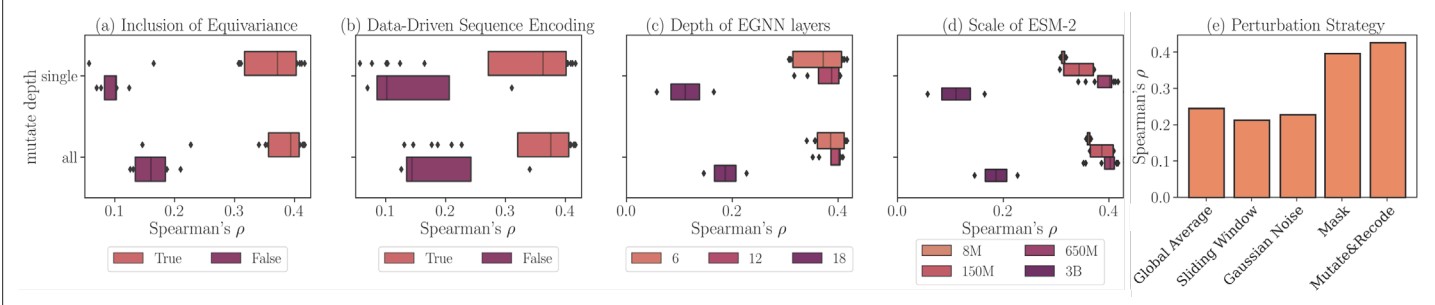

**Figure 3.** Ablation study on **ProteinGym v0** with different modular settings of ProtSSN. Each record represents the average Spearman's correlation of all assays. (**a**) Performance using different structure encoders: Equivariant Graph Neural Networks (EGNN) (orange) versus GCN/GAT (purple). (**b**) Performance using different node attributes: ESM2-embedded hidden representation (orange) versus one-hot encoding (purple). (**c**) Performance with varying numbers of EGNN layers. (**d**) Performance with different versions of ESM2 for sequence encoding. (**e**) Performance using different amino acid perturbation strategies during pre-training.

## Ablation study

The effectiveness of ProtSSN with different modular designs is examined by its performance on the early released version **ProteinGym v0** (**Notin et al., 2022a**) with 86 DMS assays. For the ablation study, the results are summarized in **Figure 3a–d**, where each record is subject to a combination of key arguments under investigation. For instance, in the top orange box of **Figure 3a**, we report all ablation results that utilize six EGNN layers for graph convolution, regardless of the different scales of ESM2 or the definitions of node attributes. For all modules investigated in this section, we separately discuss their influence on predicting mutation effects when modifying a single site or an arbitrary number of sites. The two cases are marked on the y-axis by 'single' and 'all', respectively.

### Inclusion of roto-translation equivariance

We assess the effect of incorporating rotation and translation equivariance for protein geometric and topological encoding. Three types of graph convolutions are compared, including GCN (**Kipf and Welling, 2017**), GAT (**Veličković et al., 2018**), and EGNN (**Satorras et al., 2021**). The first two are classic non-equivariant graph convolutional methods, and the last one, which we apply in the main algorithm, preserves roto-translation equivariance. We fix the number of EGNN layers to 6 and examine the performance of the other two methods with either 4 or 6 layers. We find that integrating equivariance when embedding protein geometry would significantly improve prediction performance.

### Sequence encoding

We next investigate the benefits of defining data-driven node attributes for protein representation learning. We compare the performance of models trained on two sets of graph inputs: the first set defines its AA node attributes through trained ESM2 (**Lin et al., 2023**), while the second set uses one-hot encoded AAs for each node. A clear advantage of using hidden representations by prefix models over hardcoded attributes is evident from the results presented in **Figure 3b**.

### Depth of EGNN

Although graph neural networks can extract topological information from geometric inputs, it is vital to select an appropriate number of layers for the module to deliver the most expressive node representation without encountering the oversmoothing problem. We investigate a wide range of choices for the number of EGNN layers among $\{6, 12, 18\}$. As reported in **Figure 3c**, embedding graph topology with deeper networks does not lead to performance improvements. A moderate choice of six EGNN layers is sufficient for our learning task.

### Scale of ESM

We also evaluate ProtSSN on different choices of language embedding dimensions to study the trade-off between computational cost and input richness. Various scales of prefix models, including

{8, 150, 650, 3000} millions of parameters, have been applied to produce different sequential embeddings with {320, 640, 1280, 2560} dimensions, respectively. *Figure 3d* reveals a clear preference for ESM2-t33, which employs 650 million parameters to achieve optimal model performance with the best stability. Notably, a higher dimension and richer semantic expression do not always yield better performance. A performance degradation is observed at ESM2 with 3 billion parameters.

## AA perturbation strategies

ProtSSN utilizes noise sampling at each epoch on the node attributes to emulate random mutations in nature. This introduction of noise directly affects the node attribute input to the graphs. Alongside the 'mutate-then-recode' method we implemented in the main algorithm, we examined four additional strategies to perturb the input data during training. The construction of these additional strategies is detailed below, and their corresponding Spearman's $\rho$ on **ProteinGym v0** is given in *Figure 3e*.

### Global average

Suppose the encoded sequential representation of a node is predominantly determined by its residue. In essence, the protein sequence encoder will return similar embeddings for nodes of the same residue, albeit at different positions within a protein. With this premise, the first strategy replaces the node embedding for perturbed nodes with the average representations of the same residues. For example, when perturbing an AA to glycine, an overall representation of glycine is assigned by extracting and average-pooling all glycine representations from the input sequence.

### Sliding window

The presumption in the previous method neither aligns with the algorithmic design nor biological heuristics. Self-attention discerns the interaction of the central token with its neighbors across the entire document (protein). The representation of a residue is influenced by both its neighbors' residues and locations. Thus, averaging embeddings of residues from varying positions is likely to forfeit positional information of the modified residue. For this purpose, we consider a sliding window of size 3 along the protein sequence.

**Table 2.** Influence of folding strategies (AlphaFold2 and ESMFold) on prediction performance for structure-involved models.

| | DTm | | | DDG | | | ProteinGym-interaction | | | ProteinGym-catalysis | | |
|---|---|---|---|---|---|---|---|---|---|---|---|---|
| | AlphaFold2 | ESMFold | diff ↓ | AlphaFold2 | ESMFold | **diff↓** | AlphaFold2 | ESMFold | **diff↓** | AlphaFold2 | ESMFold | **diff↓** |
| Avg. pLDDT | 90.86 | 83.22 | - | 95.19 | 86.03 | - | 82.86 | 65.69 | - | 85.80 | 73.45 | - |
| ESM-if1 | 0.395 | 0.371 | 0.024 | 0.457 | 0.488 | –0.031 | 0.351 | 0.259 | 0.092 | 0.386 | 0.368 | 0.018 |
| MIF-ST | 0.400 | 0.378 | 0.022 | 0.438 | 0.423 | –0.015 | 0.390 | 0.327 | 0.063 | 0.408 | 0.388 | 0.020 |
| k30_h1280 | 0.384 | 0.370 | 0.014 | 0.396 | 0.390 | 0.006 | 0.398 | 0.373 | 0.025 | 0.443 | 0.439 | 0.004 |
| k30_h768 | 0.359 | 0.356 | 0.003 | 0.378 | 0.366 | 0.012 | 0.400 | 0.374 | 0.026 | 0.442 | 0.436 | **0.006** |
| k30_h512 | 0.413 | 0.399 | 0.014 | 0.408 | 0.394 | 0.014 | 0.400 | 0.372 | **0.028** | 0.447 | 0.442 | 0.005 |
| k20_h1280 | 0.415 | 0.391 | 0.024 | 0.429 | 0.410 | 0.019 | 0.399 | 0.365 | 0.034 | 0.446 | 0.441 | 0.005 |
| k20_h768 | 0.415 | 0.403 | **0.012** | 0.419 | 0.397 | 0.022 | 0.401 | 0.370 | 0.031 | 0.449 | 0.442 | 0.007 |
| k20_h512 | 0.419 | 0.395 | 0.024 | 0.441 | 0.432 | 0.009 | 0.406 | 0.371 | 0.035 | 0.449 | 0.439 | 0.010 |
| k10_h1280 | 0.406 | 0.391 | 0.015 | 0.426 | 0.411 | 0.015 | 0.396 | 0.365 | 0.031 | 0.440 | 0.434 | **0.006** |
| k10_h768 | 0.400 | 0.391 | 0.009 | 0.414 | 0.400 | 0.014 | 0.379 | 0.349 | 0.030 | 0.431 | 0.421 | 0.010 |
| k10_h512 | 0.383 | 0.364 | 0.019 | 0.424 | 0.414 | **0.010** | 0.389 | 0.364 | 0.025 | 0.440 | 0.432 | 0.008 |

The top three are highlighted by first, second, and **third**.

## Gaussian noise

The third option regards node embeddings of AA as hidden vectors and imposes white noise on the vector values. We define the mean = 0 and variance = 0.5 on the noise, making the revised node representation $\tilde{v} = v + \mathcal{N}(0, 0.5)$.

## Mask

Finally, we employ the masking technique prevalent in masked language modeling and substitute the perturbed residue token with a special `<mask>` token. The prefix language model will interpret it as a novel token and employ self-attention mechanisms to assign a new representation to it. We utilize the same hyperparameter settings as that of BERT (*Devlin et al., 2018*) and choose 15% of tokens per iteration as potentially influenced subjects. Specifically, 80% of these subjects are replaced with `<mask>`, 10% of them are replaced randomly (to other residues), and the remaining 10% stay unchanged.

## Folding methods

The analysis of protein structure through topological embeddings poses challenges due to the inherent difficulty in obtaining accurate structural observations through experimental techniques such as NMR (*Wüthrich, 1990*) and cryo-EM (*Elmlund et al., 2017*). As a substitution, we use in silico folding models such as AlphaFold2 (*Jumper et al., 2021*) and ESMFold (*Lin et al., 2023*). Although these folding methods may introduce additional errors due to the inherent uncertainty or inaccuracy in structural predictions, we investigate the robustness of our model in the face of potential inaccuracies introduced by these folding strategies.

*Table 2* examines the impact of different folding strategies on the performance of DMS predictions for ESM-if1, MIF-ST, and ProtSSN. Although the performance based on ESMFold-derived protein structures generally lags behind structures folded by AlphaFold2, the minimal difference between the two strategies across all four benchmarks validates the robustness of our ProtSSN in response to variations in input protein structures. We deliberately divide the assays in **ProteinGym v0** (*Notin et al., 2022a*) into two groups of interaction (e.g., binding and stability) and catalysis (e.g., activity), as the former is believed more sensitive to the structure of the protein (*Robertson and Murphy, 1997*).

ProtSSN emerges as the most robust method, maintaining consistent performance across different folding strategies, which underscores the importance of our model's resilience to variations in input protein structures. The reported performance difference for both ESM-if1 and MIF-ST is 3–5 times higher than that of ProtSSN. The inconsistency between the optimal results for DDG and the scores reported in *Table 1* comes from the utilization of crystal structures in the main results. Another noteworthy observation pertains to the performance of ESM-if1 and MIF-ST on DDG. In this case, predictions based on ESMFold surpass those based on AlphaFold2, even outperforming predictions derived from crystal structures. However, the superior performance of these methods with ESMFold hinges on inaccurate protein structures, rendering them less reliable.

## Performance enhancement with MSA and ensemble

We extend the evaluation on **ProteinGym v0** to include a comparison of our individual-sequence-level, zero-shot ProtSSN with methods that include MSAs (which we consider here, 'few-shot' methods [*Meier et al., 2021*]). The details are reported in *Table 3*, where the performance of baseline methods is retrieved from https://github.com/OATML-Markslab/ProteinGym, copy archived at *ProteinGym, 2024*. For ProtSSN, we employ an ensemble of nine models with varying sizes of $k$NN graphs ($k \in \{10, 20, 30\}$) and EGNN hidden neurons ($\{512, 768, 1280\}$), as discussed previously. The MSA depth reflects the number of MSA sequences that can be found for the protein, which influences the quality of MSA-based methods.

ProtSSN demonstrates superior performance among non-MSA zero-shot methods in predicting both single-site and multi-site mutants, underscoring its pivotal role in guiding protein engineering toward deep mutation. Furthermore, ProtSSN outperforms MSA-based methods across taxa, except for viruses, where precise identification of conserved sites is crucial for positive mutations, especially in spike proteins (*Katoh and Standley, 2021*; *Mercurio et al., 2021*). In essence, ProtSSN provides an efficient and effective solution for variant effects prediction when compared to both non-MSA and MSA-based methods. Note that although MSA-based methods generally consume fewer trainable

**Table 3.** Variant effect prediction on **ProteinGym v0** with both zero-shot and few-shot methods. Results are retrieved from **Notin et al., 2022a**.

| Model | Type | # Params (million) | ρ (by mutation depth)↑ | | | ρ (by MSA depth)↑ | | | ρ (by taxon)↑ | | | |
|---|---|---|---|---|---|---|---|---|---|---|---|---|
| | | | Single | Double | All | Low | Medium | High | Prokaryote | Human | Eukaryote | Virus |
| *Few-shot methods* | | | | | | | | | | | | |
| SiteIndep | Single | - | 0.378 | 0.322 | 0.378 | 0.431 | 0.375 | 0.342 | 0.343 | 0.375 | 0.401 | 0.406 |
| EVmutation | Single | - | 0.423 | 0.401 | 0.423 | 0.406 | 0.403 | 0.484 | 0.499 | 0.396 | 0.429 | 0.381 |
| Wavenet | Single | - | 0.399 | 0.344 | 0.400 | 0.286 | 0.404 | 0.489 | 0.492 | 0.373 | 0.442 | 0.321 |
| GEMME | Single | - | 0.460 | 0.397 | 0.463 | 0.444 | 0.446 | 0.520 | 0.505 | 0.436 | 0.479 | 0.451 |
| | Single | - | 0.411 | 0.358 | 0.416 | 0.386 | 0.391 | 0.505 | 0.497 | 0.396 | 0.461 | 0.332 |
| DeepSequence | Ensemble | - | 0.433 | 0.394 | 0.435 | 0.386 | 0.411 | 0.534 | 0.522 | 0.405 | 0.480 | 0.361 |
| | Single | - | 0.451 | 0.406 | 0.452 | 0.417 | 0.434 | 0.525 | 0.518 | 0.411 | 0.469 | 0.436 |
| EVE | Ensemble | - | 0.459 | 0.409 | 0.459 | 0.424 | 0.441 | 0.532 | 0.526 | 0.419 | 0.481 | 0.437 |
| | Single | 100 | 0.405 | 0.358 | 0.426 | 0.372 | 0.415 | 0.500 | 0.506 | 0.387 | 0.468 | 0.379 |
| MSA-Transfomer | Ensemble | 500 | 0.440 | 0.374 | 0.440 | 0.387 | 0.428 | 0.513 | 0.517 | 0.398 | 0.467 | 0.406 |
| Tranception-R | Single | 700 | 0.450 | 0.427 | 0.453 | 0.442 | 0.438 | 0.500 | 0.495 | 0.424 | 0.485 | 0.434 |
| TranceptEVE | Single | 700 | 0.481 | 0.445 | 0.481 | 0.454 | 0.465 | 0.542 | 0.539 | 0.447 | 0.498 | 0.461 |
| *Zero-shot methods* | | | | | | | | | | | | |
| RITA | Ensemble | 2,210 | 0.393 | 0.236 | 0.399 | 0.350 | 0.414 | 0.407 | 0.391 | 0.391 | 0.405 | 0.417 |
| ESM-1v | Ensemble | 3,250 | 0.416 | 0.309 | 0.417 | 0.390 | 0.378 | 0.536 | 0.521 | 0.439 | 0.423 | 0.268 |
| ProGen2 | Ensemble | 10,779 | 0.421 | 0.312 | 0.423 | 0.384 | 0.421 | 0.467 | 0.497 | 0.412 | 0.459 | 0.373 |
| ProtTrans | Ensemble | 6,840 | 0.417 | 0.360 | 0.413 | 0.372 | 0.395 | 0.492 | 0.498 | 0.419 | 0.400 | 0.322 |
| ESM2 | Ensemble | 18,843 | 0.415 | 0.316 | 0.413 | 0.391 | 0.381 | 0.509 | 0.508 | 0.456 | 0.461 | 0.213 |
| SaProt | Ensemble | 1,285 | 0.447 | 0.368 | 0.450 | 0.456 | 0.410 | 0.544 | 0.534 | 0.464 | 0.460 | 0.334 |

*Table 3 continued on next page*

*Table 3 continued*

| Model | Type | # Params (million) | ρ (by mutation depth)↑ | | | ρ (by MSA depth)↑ | | | ρ (by taxon)↑ | | | |
|---|---|---|---|---|---|---|---|---|---|---|---|---|
| | | | Single | Double | All | Low | Medium | High | Prokaryote | Human | Eukaryote | Virus |
| ProtSSN | Ensemble | 1,476 | 0.433 | 0.381 | 0.433 | 0.406 | 0.402 | 0.532 | **0.530** | 0.436 | 0.491 | 0.290 |

The top three are highlighted by first, second, and third.

parameters than non-MSA methods, they incur significantly higher costs in terms of search time and memory usage. Moreover, the inference performance of MSA-based methods relies heavily on the quality of the input MSA, where the additionally involved variance makes impacts the stability of the model performance.

## Discussion

The development of modern computational methodologies for protein engineering is a crucial facet of in silico protein design. Effectively assessing the fitness of protein mutants not only supports cost-efficient experimental validations but also guides the modification of proteins toward enhanced or novel functions. Traditionally, computational methods rely on analyzing small sets of labeled data for specific protein assays, such as FLIP (*Dallago et al., 2021*), PEER (*Xu et al., 2022*), and PETA (*Tan et al., 2024b*). More recent work has also focused on examining the relationship between models and supervised fitness prediction (*Li et al., 2024*). However, obtaining experimental data is often infeasible in real-world scenarios, particularly for positive mutants, due to the cost and complexity of protein engineering. This limitation renders supervised learning methods impractical for many applications. As a result, it is crucial to develop solutions that can efficiently suggest site mutations for wet experiments, even when starting from scratch without prior knowledge. Recent deep learning solutions employ a common strategy that involves establishing a hidden protein representation and masking potential mutation sites to recommend plausible AAs. Previous research has primarily focused on extracting protein representations from either their sequential or structural modalities, with many treating the prediction of mutational effects merely as a secondary task following inverse folding or de novo protein design. These approaches often overlook the importance of comprehensive investigation on both global and local perspectives of AA interaction that are critical for featuring protein functions. Furthermore, these methods hardly tailored model designs for suggesting mutations, despite the significance of this type of task. In this work, we introduce ProtSSN, a denoising framework that effectively cascades protein sequence and structure embedding for predicting mutational effects. Both the protein language model and equivariant graph neural network are employed to encode the global interaction of AAs with geometry-aware local enhancements.

On the other hand, existing benchmarks for evaluating computational solutions mostly focus on assessing model generalization on large-scale datasets (e.g., over 2 million mutants in **ProteinGym v1**). However, such high-throughput datasets often lack accurate experimental measurements (*Frazer et al., 2021*), leading to significant noise in the labels. Furthermore, the larger data volumes typically do not include environmental labels for mutants (e.g., temperature, pH), despite the critical importance of these conditions for biologists. In response, we propose **DTm** and **DDG**. These low-throughput datasets, which we have collected, emphasize the consistency of experimental conditions and data accuracy. They are traceable and serve as valuable complements to ProteinGym, providing a more detailed and reliable evaluation of computational models. We have extensively validated the efficacy of ProtSSN across various protein function assays and taxa, including two thermostability

**Table 4.** Average Spearman's $\rho$ correlation of variant effect prediction on **DTm** and **DDG** for zero-shot methods with model ensemble.
The values within () indicate the standard deviation of bootstrapping.

| Model | # Params (million) | DTm | DDG |
|---|---|---|---|
| RITA | 2210 | $0,195_{(0.045)}$ | $0.255_{(0.061)}$ |
| ESM-1v | 3250 | $0.300_{(0.036)}$ | $0.310_{(0.054)}$ |
| Tranception | 1085 | $0.202_{(0.039)}$ | $0.277_{(0.062)}$ |
| ProGen2 | 10,779 | $0.293_{(0.042)}$ | $0.282_{(0.063)}$ |
| ProtTrans | 6840 | $0.323_{(0.039)}$ | $0.389_{(0.059)}$ |
| ESM2 | 18,843 | $0.346_{(0.035)}$ | $0.383_{(0.55)}$ |
| SaProt | 1285 | $0.392_{(0.040)}$ | $0.415_{(0.061)}$ |
| ProtSSN | 1476 | $\mathbf{0.425}_{(0.033)}$ | $\mathbf{0.440}_{(0.057)}$ |

**Table 5.** Statistical summary of **DTm** and **DDG**.

| pH range | DTm | | | DDG | | |
|---|---|---|---|---|---|---|
| | # Assays | Avg. # mut | Avg. AA | # Assays | Avg. # mut | Avg.AA |
| Acid (pH < 7) | 29 | 29.1 | 272.8 | 21 | 22.6 | 125.3 |
| Neutral (pH = 7) | 14 | 37.4 | 221.2 | 10 | 23.1 | 78.3 |
| Alkaline (pH > 7) | 23 | 50.1 | 233.3 | 5 | 24.6 | 101.4 |
| Sum | 66 | 38.2 | 221.2 | 36 | 23.0 | 108.8 |

datasets prepared by ourselves. Our approach consistently demonstrates substantial promise for protein engineering applications, such as facilitating the design of mutation sequences with enhanced interaction and enzymatic activity (*Table 4*).

# Materials and methods

This section starts by introducing the three mutational scanning benchmark datasets used in this study, including an open benchmark dataset **ProteinGym v1** and two new datasets on protein thermostability, that is, **DTm** and **DDG**. We establish the proposed ProtSSN for protein sequences and structures encoding in the section 'Proposed method'. The overall pipeline of model training and inference is presented in the section 'Model pipeline'. The experimental setup for model evaluation is provided in the section 'Experimantal protocol'.

**Table 6.** Details of zero-shot baseline models.

| Model | Type | | Description | Source code |
|---|---|---|---|---|
| | Seq | Struct | | |
| RITA (*Hesslow et al., 2022*) | ✓ | | A generative protein language model with billion-level parameters | https://github.com/lightonai/RITA (*Hesslow and Poli, 2023*) |
| ProGen2 (*Nijkamp et al., 2023*) | ✓ | | A generative protein language model with billion-level parameters | https://github.com/salesforce/progen (*Nijkamp, 2022*) |
| ProtTrans (*Elnaggar et al., 2021*) | ✓ | | Transformer-based models trained on large protein sequence corpus | https://github.com/agemagician/ProtTrans (*Elnaggar and Heinzinger, 2025*) |
| Tranception (*Notin et al., 2022a*) | ✓ | | An autoregressive model for variant effect prediction with retrieve machine | https://github.com/OATML-Markslab/Tranception (*Notin, 2023*) |
| CARP (*Yang et al., 2024*) | ✓ | | Pretrained CNN protein sequence masked language models of various sizes | https://github.com/microsoft/protein-sequence-models (*microsoft, 2024*) |
| ESM-1b (*Rives et al., 2021*) | ✓ | | A masked language model-based pre-train method with various pre-training dataset and positional embedding strategies | https://github.com/facebookresearch/esm (*facebookresearch, 2023*) |
| ESM-1v (*Meier et al., 2021*) | ✓ | | | |
| ESM2 (*Lin et al., 2023*) | ✓ | | | |
| ESM-if1 (*Hsu et al., 2022*) | | ✓ | An inverse folding method with mask language modeling and geometric vector perception (*Jing et al., 2020*) | |
| MIF-ST (*Yang et al., 2023*) | ✓ | ✓ | Pretrained masked inverse folding models with sequence pretraining transfer | https://github.com/microsoft/protein-sequence-models |
| SaProt (*Su et al., 2023*) | ✓ | ✓ | Structure-aware vocabulary for protein language modeling with FoldSeek | https://github.com/westlake-repl/SaProt (*Su and fajieyuan, 2025*) |

## Dataset

To evaluate model performance in predicting mutation effects, we compare ProtSSN with SOTA baselines on **ProteinGym v1**, the largest open benchmark for DMS assays. We also introduce two novel benchmarks, **DTm** and **DDG**, for assessing protein thermostability under consistent control environments. Details of the new datasets are provided in *Table 5*. Considering the significant influence of experimental conditions on protein temperature, we explicitly note the pH environment in each assay. Refer to the following paragraphs for more details.

## ProteinGym

The assessment of ProtSSN for deep mutation effects prediction is conducted on **ProteinGym v1** (*Notin et al., 2023*). It is the most extensive protein substitution benchmark comprising 217 assays and more than 2.5*M* mutants. These DMS assays cover a broad spectrum of functional properties (e.g., ligand binding, aggregation, viral replication, and drug resistance) and span various protein families (e.g., kinases, ion channel proteins, transcription factors, and tumor suppressors) across different taxa (e.g., humans, other eukaryotes, prokaryotes, and viruses) (*Table 6*).

## DTm

The assays in the first novel thermostability benchmark are sourced from single-site mutations in **ProThermDB** (*Nikam et al., 2021*). Each assay is named in the format '*UniProt_ID-pH*'. For instance, '*O00095-8.0*' signifies mutations conducted and evaluated under pH 8.0 for protein O00095. The attributes include 'mutant', 'score', and 'UniProt_ID', with at least 10 mutants to ensure a meaningful evaluation. To concentrate on single-site mutations compared to wild-type proteins, we exclude records with continuous mutations. To avoid dealing with partial or misaligned protein structures resulting from incomplete wet experimental procedures, we employ UniProt ID to pair protein sequences with folded structures predicted by AlphaFold2 (*Jumper et al., 2021*) from https://alphafold.ebi.ac.uk. In total, **DTm** consists of 66 such protein–environment pairs and 2520 mutations.

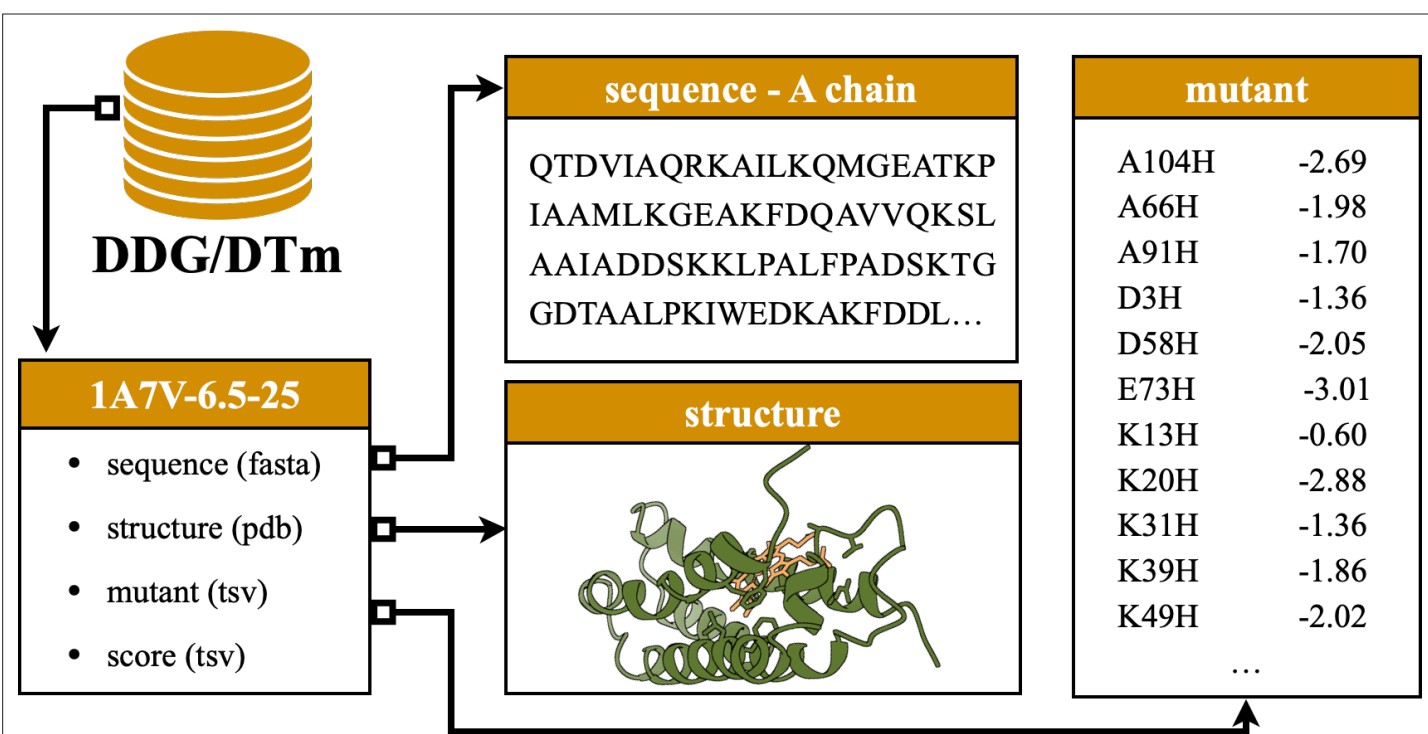

**Figure 4.** An example source record of the mutation assay. The record is derived from DDG for the A chain of protein 1A7V, experimented at pH = 6.5 and degree at 25°C.

## DDG

The second novel thermostability benchmark is sourced from **DDMut** (*Zhou et al., 2023*), where mutants are paired with their PDB documents. We thus employ crystal structures for proteins in DDG and conduct additional preprocessing steps for environmental consistency. In particular, we group mutants of the same protein under identical experimental conditions and examine the alignment between mutation sites and sequences extracted from crystal structures. We discard mutant records that cannot be aligned with the sequence loaded from the associated PDB document. After removing data points with fewer than 10 mutations, **DDG** comprises 36 data points and 829 mutations. As before, assays are named in the format '*PDB_ID-pH-temperature*' to indicate the chemical condition (pH) and temperature (in Celsius) of the experiment on the protein. An example assay content is provided in *Figure 4*.

## Proposed method

Labeled data are usually scarce in biomolecule research, which demands designing a general self-supervised model for predicting variant effects on unknown proteins and protein functions. We propose ProtSSN with a pre-training scheme that recovers residues from a given protein backbone structure with noisy local environment observations.

## Multilevel protein representation

### Protein primary structure (Noisy)

Denote a given observation with residue $\tilde{v}$. We assume this arbitrary observation is undergoing random mutation toward a stable state. The training objective is to learn a revised state $v$ that is less likely to be eliminated by natural selection due to unfavorable properties such as instability or inability to fold. The perturbed observation is defined by a multinomial distribution.

$$\pi(\tilde{v} \mid v) = p\Theta(\pi_1, \pi_2, \ldots, \pi_{20}) + (1 - p)\delta(\tilde{v} - v), \tag{1}$$

where an AA in a protein chain has a chance of $p$ to mutate to one of 20 AAs (including itself) following a *replacement distribution* $\Theta(\cdot)$ and $(1 - p)$ of remaining unchanged. We consider $p$ as a tunable parameter and define $\Theta(\cdot)$ by the frequency of residues observed in wild-type proteins.

### Protein tertiary structure

The geometry of a protein is described by $\mathcal{G} = (\mathcal{V}, \mathcal{E}, W_V, W_E, X_V)$, which is a residue graph based on the $k$-nearest neighbor algorithm ($k$NN). Each node $v_i \in \mathcal{V}$ represents an AA in the protein connected to up to k closest nodes within 30 Å. Node attributes $W_V$ are hidden semantic embeddings of residues extracted by the semantic encoder (see the section 'Semantic encoding of global AA contacts), and edge attributes $W_E \in \mathbb{R}^{93}$ feature relationships of connected nodes based on inter-atomic distances, local N-C positions, and sequential position encoding (*Zhou et al., 2024b*). Additionally, $X_V$ records 3D coordinates of AAs in the Euclidean space, which plays a crucial role in the subsequent geometric embedding stage to preserve roto-translation equivariance.

## Semantic encoding of global AA contacts

Although it is generally believed by the community that a protein's sequence determines its biological function via the folded structure, following strictly to this singular pathway risks overlooking other unobserved yet potentially influential inter-atomic communications impacting protein fitness. Our proposed ProtSSN thus includes pairwise relationships for residues through an analysis of proteins' primary structure from $\tilde{V}$ and embeds them into hidden representations $W_V$ for residues. At each iteration, the input sequences are modified randomly by *Equation 3* and then embedded via an *Evolutionary Scale Modeling* method, ESM2 (*Lin et al., 2023*), which employs a BERT-style masked language modeling objective. This objective predicts the identity of randomly selected AAs in a protein sequence by observing their context within the remainder of the sequence. Alternative semantic encoding strategies are discussed in the section 'AA perturbation strategies'.

## Geometrical encoding of local AA contacts

Proteins are structured in 3D space, which requires the geometric encoder to possess roto-translation equivariance to node positions as well as permutation invariant to node attributes. This design is vital

to avoid the implementation of costly data augmentation strategies. We practice EGNN (*Satorras et al., 2021*) to acquire the hidden node representation $W_V^{l+1} = \left\{ w_{v_1}^{l+1}, \ldots, w_{v_n}^{l+1} \right\}$ and node coordinates $X_{\text{pos}}^{l+1} = \left\{ x_{v_1}^{l+1}, \ldots, x_{v_n}^{l+1} \right\}$ at the $l$+1th layer

$$
\begin{aligned}
m_{ij} &= \phi_e \left( w_{v_i}^l, w_{v_j}^l, \left\| x_{v_i}^l - x_{v_j}^l \right\|^2, w_{e_{ij}} \right), \\
x_{v_i}^{l+1} &= \mathbf{x}_{v_i}^l + \frac{1}{n} \sum_{j \neq i} \left( \mathbf{x}_{v_i}^l - \mathbf{x}_{v_j}^l \right) \phi_x \left( m_{ij} \right), \\
w_{v_i}^{l+1} &= \phi_v \left( w_i^l, \sum_{j \neq i} m_{ij} \right).
\end{aligned}
\tag{2}
$$

In these equations, $w_{e_{ij}}$ represents the input edge attribute on $\mathcal{V}_{ij}$, which is not updated by the network. The propagation rules $\phi_e, \phi_x,$ and $\phi_v$ are defined by differentiable functions, for example, .multilayer perceptrons (MLPs). The final hidden representation on nodes $W_V^L$ embeds the microenvironment and local topology of AAs, and it will be carried on by readout layers for label predictions.

## Model pipeline

ProtSSN is designed for protein engineering with a self-supervised learning scheme. The model is capable of conducting zero-shot variant effect prediction on an unknown protein, and it can generate the joint distribution for the residue of all AAs along the protein sequence of interest. This process accounts for the epistatic effect and concurrently returns all AA sites in a sequence. Below, we detail the workflow for training the zero-shot model and scoring the effect of a specific mutant.

### Training

The fundamental model architecture cascades a frozen sequence encoding module and a trainable tertiary structure encoder. The protein is represented by its sequence and structure characteristics, where the former is treated as plain text and the latter is formalized as a $k$NN graph with model-encoded node attributes, handcrafted edge attributes, and 3D positions of the corresponding AAs. Each input sequence $V$ are first one-hot encoded by 33 tokens, comprising both AA residues and special tokens (*Lin et al., 2023*). The ground truth sequences, during training, will be permuted into

$$
\tilde{V} = Permute(V).
\tag{3}
$$

Here, $\tilde{V}$ is the perturbed AA to be encoded by a protein language model, which analyses pairwise hidden relationships among AAs from the input protein sequence and produces a vector representation $w_{v_i} \in W_V$ for each AA, that is,

$$
W_V = \text{LM}_{\text{frozen}}(\tilde{V})
\tag{4}
$$

The language model $\text{LM}_{\text{frozen}}(\cdot)$, ESM2 (*Lin et al., 2023*) for instance, has been pre-trained on a massive protein sequence database (e.g., UniRef50 [*Suzek et al., 2015*]) to understand the semantic and grammatical rules of wild-type proteins with high-dimensional AA-level long-short-range interactions. The independent perturbation on AA residues follows (*Equation 1*) operates in every epoch, which requires constant updates on the sequence embedding. To further enhance the interaction of locally connected AAs in the node representation, a stack of LEGNN layers is implemented on the input graph $\mathcal{G}$ to yield

$$
W_V^L = \text{EGNN}(\mathcal{G}).
\tag{5}
$$

During the pre-training phase for protein sequence recovery, the output layer $\phi(\cdot)$ provides the probability of tokens in one of 33 types, that is,

$$
Y = \phi(W_V^L) \in \mathbb{R}^{n \times 33}
\tag{6}
$$

for a protein of $n$ AAs. The model's learnable parameters are refined by minimizing the cross-entropy of the recovered AAs with respect to the ground-truth AAs in wild-type proteins:

$$
\text{loss} = -\sum_{n=1}^{L} \sum_{c=1}^{33} V_{n,c} \log \left( \text{softmax}(Y_{n,c}) \right),
\tag{7}
$$

where $L$ is the length of the sequence, $n$ represents the position index within the sequence, and each $c$ corresponds to a potential type of token.

## Inferencing

For a given wild-type protein and a set of mutants, the fitness scores of the mutants are derived from the joint distribution of the altered residues relative to the wild-type template. First, the wild-type protein sequence and structure are encoded into hidden representations following (*Equations 4 and 5*). Unlike the pre-training process, here the residue in the wild-type protein is considered as a reference state and is compared with the predicted probability of AAs at the mutated site. Next, for a mutant with mutated sites $\mathcal{T}$ ($|\mathcal{T}| \geq 1$), we define its fitness score $F_x$ using the corresponding *log-odds ratio*, that is,

$$F_x = \sum_{t \in \mathcal{T}} \log p(\mathbf{y}_t) - \log p(\mathbf{v}_t), \tag{8}$$

where $\mathbf{y}_t$ and $\mathbf{v}_t$ denote the mutated and wild-type residue of the $t$th AA.

## Experimental protocol

We validate the efficacy of ProtSSN on zero-shot mutation effect prediction tasks. The performance is compared with other SOTA models of varying numbers of trainable parameters. The implementations are programmed with `PyTorch-Geometric` (ver 2.2.0) and `PyTorch` (ver 1.12.1) and executed on an NVIDIA Tesla A100 GPU with 6912 CUDA cores and 80GB HBM2 installed on an HPC cluster.

## Training setup

ProtSSN is pre-trained on a non-redundant subset of **CATH v4.3.0** (*Orengo et al., 1997*) domains, which contains 30,948 experimental protein structures with less than 40% sequence identity. We remove proteins that exceed 2000 AAs in length for efficiency. For each $k$NN protein graph, node features are extracted and updated by a frozen ESM2-t33 prefix model (*Lin et al., 2023*). Protein topology is inferred by a six-layer EGNN (*Satorras et al., 2021*) with the hidden dimension tuned from $\{512, 768, 1280\}$. Adam (*Kingma and Ba, 2015*) is used for backpropagation with the learning rate set to 0.0001. To avoid training instability or CUDA out-of-memory, we limit the maximum input to 8192 AA tokens per batch, constituting approximately 32 graphs.

## Baseline methods

We undertake an extensive comparison with baseline methods of self-supervised sequence or structure-based models on the fitness of mutation effects prediction (*Table 5*). Sequence models employ position embedding strategies such as autoregression (RITA [*Hesslow et al., 2022*], Tranception [*Notin et al., 2022a*], TranceptEVE [*Notin et al., 2022b*], and ProGen2 [*Nijkamp et al., 2023*]), masked language modeling (ESM-1b [*Rives et al., 2021*], ESM-1v [*Meier et al., 2021*], and ESM2 [*Lin et al., 2023*]), a combination of both (ProtTrans [*Elnaggar et al., 2021*]), and convolutional modeling (CARP [*Yang et al., 2024*]). Additionally, we also compare with ESM-if1 (*Hsu et al., 2022*) which incorporates masked language modeling objectives with GVP (*Jing et al., 2020*), as well as MIF-ST (*Yang et al., 2023*) and SaProt (*Su et al., 2023*) with both sequence and structure encoding.

The majority of the performance comparison was conducted on zero-shot deep learning methods. However, for completeness, we also report a comparison with popular MSA-based methods, such as GEMME (*Laine et al., 2019*), Site-Independent (*Hopf et al., 2017*), EVmutation (*Hopf et al., 2017*), Wavenet (*Shin et al., 2021*), DeepSequence (*Riesselman et al., 2018*), and EVE (*Frazer et al., 2021*). Other deep learning methods use MSA for training or retrieval, including MSA-Transformer (*Rao et al., 2021b*), Tranception (*Notin et al., 2022a*), and TranceptEVE (*Notin et al., 2022b*).

## Scoring function

Following the convention, the fitness scores of the zero-shot models are calculated by the log-odds ratio in *Equation 8* for encoder methods, such as the proposed ProtSSN and the ProtTrans series models. For autoregressive and inverse folding models (e.g., Tranception and ProGen2) that predict the next token $x_i$ based on the context of previous $x_{1:i-1}$ tokens, the fitness score $F_x$ of a mutated sequence $\mathbf{y}$ is computed via the log-likelihood ratio with the wild-type sequence $\mathbf{v}$, that is,

$$F_x = \log \frac{P(\boldsymbol{y})}{P(\boldsymbol{v})}. \tag{9}$$

## Benchmark datasets

We conduct a comprehensive comparison of deep learning predictions on hundreds of DMS assays concerning different protein types and biochemical assays. We prioritize experimentally measured properties that possess a monotonic relationship with protein fitness, such as catalytic activity, binding affinity, expression, and thermostability. In particular, **ProteinGym v1** (*Notin et al., 2023*) groups five sets of protein properties from 217 assays. Two new datasets named **DTm** and **DDG** examine 102 environment-specific assays with thermostability scores. In total, 319 assays with around 2.5 million mutants are scored. To better understand the novelty of the proposed ProtSSN, we designed additional investigations on **ProteinGym v0** (*Notin et al., 2022a*), an early-release version of **ProteinGym** with $1.5M$ missense variants across 86 assays (excluding one assay that failed to be folded by both AlphaFold2 and ESMFold, that is, `A0A140D2T1_ZIKV_Sourisseau_growth_2019`).

## Evaluation metric

We evaluate the performance of pre-trained models on a diverse set of proteins and protein functions using Spearman's $\rho$ correlation that measures the strength and direction of the monotonic relationship between two ranked sequences, that is, experimentally evaluated mutants and model-inferred mutants (*Meier et al., 2021*; *Notin et al., 2022a*; *Frazer et al., 2021*; *Zhou et al., 2024b*). This non-parametric rank measure is robust to outliers and asymmetry in mutation scores, and it does not assume any specific distribution of mutation fitness scores. The scale of $\rho$ ranges from -1 to 1, indicating the negative or positive correlation of the predicted sequence to the ground truth. The prediction is preferred if its $\rho$ to the experimentally examined ground truth is close to 1.

## Acknowledgements

This work was supported by the Science and Technology Innovation Key R&D Program of Chongqing (CSTB2022TIAD-STX0017), the National Science Foundation of China (Grant Number 62302291, 12104295), the Computational Biology Key Program of Shanghai Science and Technology Commission (23JS1400600), Shanghai Jiao Tong University Scientific and Technological Innovation Funds (21X010200843), the Student Innovation Center at Shanghai Jiao Tong University, and Shanghai Artificial Intelligence Laboratory.

## Additional information

### Funding

| Funder | Grant reference number | Author |
| --- | --- | --- |
| Science and Technology Innovation Key R&D Program of Chongqing | CSTB2022TIAD-STX0017 | Liang Hong |
| National Natural Science Foundation of China | 62302291 | Bingxin Zhou |
| National Natural Science Foundation of China | 12104295 | Liang Hong |
| The Computational Biology Key Program of Shanghai Science and Technology Commission | 23JS1400600 | Liang Hong |
| Shanghai Jiao Tong University Scientific and Technological Innovation Funds | 21X010200843 | Liang Hong |

| Funder | Grant reference number | Author |
|---|---|---|

The funders had no role in study design, data collection and interpretation, or the decision to submit the work for publication.

## Author contributions
Yang Tan, Data curation, Software, Formal analysis, Methodology; Bingxin Zhou, Conceptualization, Data curation, Supervision, Funding acquisition, Investigation, Visualization, Methodology, Writing – original draft, Project administration, Writing – review and editing, Formal analysis; Lirong Zheng, Guisheng Fan, Writing – review and editing; Liang Hong, Project administration, Writing – review and editing

## Author ORCIDs
Yang Tan http://orcid.org/0009-0004-7261-1705
Bingxin Zhou https://orcid.org/0000-0002-3897-9766
Lirong Zheng https://orcid.org/0000-0001-6803-5048
Liang Hong https://orcid.org/0000-0003-0107-336X

Reviewer #1 (Public review): https://doi.org/10.7554/eLife.98033.4.sa1
Reviewer #2 (Public review): https://doi.org/10.7554/eLife.98033.4.sa2
Author response https://doi.org/10.7554/eLife.98033.4.sa3

# Additional files

## Supplementary files
MDAR checklist

Source code 1. The source code .ZIP file contains the complete implementation of model training and evaluation, the associated processed datasets, and the .README document which provides a general instruction.

## Data availability
The source code and datasets can be found at https://github.com/ai4protein/ProtSSN (copy archived at *Tan, 2025*).

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
