## [Editor Report · eLife Assessment]

ProtSSN is a **valuable** approach that generates protein embeddings by integrating sequence and structural information, demonstrating improved prediction of mutation effects on thermostability compared to competing models. The evidence supporting the authors' claims is **compelling**, with well-executed comparisons. This work will be of particular interest to researchers in bioinformatics and structural biology, especially those focused on protein function and stability.

---

## [Referee Report · Reviewer #1 (Public review)]

Summary:

The authors introduce a denoising-style model that incorporates both structure and primary-sequence embeddings to generate richer embeddings of peptides. My understanding is that the authors use ESM for the primary sequence embeddings, take resolved structures (or use structural predictions from AlphaFold when they're not available), then develop an architecture to combine these two with a loss that seems reminiscent of diffusion models or masked language model approaches. The embeddings can be viewed as ensemble-style embedding of the two levels of sequence information, or with AlphaFold, an ensemble of two methods (ESM+AlphaFold). The authors also gather external datasets to evaluate their approach and compare it to previous approaches. The approach seems promising and appears to out-compete previous methods at several tasks. Nonetheless, I have strong concerns about a lack of verbosity as well as exclusion of relevant methods and references.

Advances:

I appreciate the breadth of the analysis and comparisons to other methods. The authors separate tasks, models, and sizes of models in an intuitive, easy-to-read fashion that I find valuable for selecting a method for embedding peptides. Moreover, the authors gather two datasets for evaluating embeddings' utility for predicting thermostability. Overall, the work should be helpful for the field as more groups choose methods/pretraining strategies amenable to their goals, and can do so in an evidence-guided manner.

Considerations:

Primarily, a majority of the results and conclusions (e.g., Table 3) are reached using data and methods from ProteinGym, yet the best-performing methods on ProteinGym are excluded from the paper (e.g., EVE-based models and GEMME). In the ProteinGym database, these methods outperform ProtSSN models. Moreover, these models were published over a year---or even 4 years in the case of GEMME---before ProtSSN, and I do not see justification for their exclusion in the text.

Secondly, related to comparison of other models, there is no section in the methods about how other models were used, or how their scores were computed. When comparing these models, I think it's crucial that there are explicit derivations or explanations for the exact task used for scoring each method. In other words, if the pre-training is indeed the important advance of the paper, the paper needs to show this more explicitly by explaining exactly which components of the model (and previous models) are used for evaluation. Are the authors extracting the final hidden layer representations of the model, treating these as features, then using these features in a regression task to predict fitness/thermostability/DDG etc.? How are the model embeddings of other methods being used, since, for example, many of these methods output a k-dimensional embedding of a given sequence, rather than one single score that can be correlated with some fitness/functional metric. Summarily, I think the text is lacking an explicit mention of how these embeddings are being summarized or used, as well as how this compares to the model presented.

I think the above issues can mainly be addressed by considering and incorporating points from Li et al. 2024[1] and potentially Tang & Koo 2024[2]. Li et al.[1] make extremely explicit the use of pretraining for downstream prediction tasks. Moreover, they benchmark pretraining strategies explicitly on thermostability (one of the main considerations in the submitted manuscript), yet there is no mention of this work nor the dataset used (FLIP (Dallago et al., 2021)) in this current work. I think a reference and discussion of [1] is critical, and I would also like to see comparisons in line with [1], as [1] is very clear about what features from pretraining are used, and how. If the comparisons with previous methods were done in this fashion, this level of detail needs to be included in the text.

To conclude, I think the manuscript would benefit substantially from a more thorough comparison of previous methods. Maybe one way of doing this is following [1] or [2], and using the final embeddings of each method for a variety of regression tasks---to really make clear where these methods are performing relative to one another. I think a more thorough methods section detailing how previous methods did their scoring is also important. Lastly, TranceptEVE (or a model comparable to it) and GEMME should also be mentioned in these results, or at the bare minimum, be given justification for their absence.

[1] Feature Reuse and Scaling: Understanding Transfer Learning with Protein Language Models, Francesca-Zhoufan Li, Ava P. Amini, Yisong Yue, Kevin K. Yang, Alex X. Lu bioRxiv 2024.02.05.578959; doi: https://doi.org/10.1101/2024.02.05.578959

[2] Evaluating the representational power of pre-trained DNA language models for regulatory genomics, Ziqi Tang, Peter K Koo bioRxiv 2024.02.29.582810; doi: https://doi.org/10.1101/2024.02.29.582810

Comments on revisions:

My concerns have been addressed. What seems to remain are some semantical disagreements and I'm not sure that these will be answered here. Do MSAs and other embedding methods lead to some notable type of data leakage? Does this leakage qualify as "x-shot" learning under current definitions?

---

## [Referee Report · Reviewer #2 (Public review)]

Summary:

To design proteins and predict disease, we want to predict the effects of mutations on the function of a protein. To make these predictions, biologists have long turned to statistical models that learn patterns that are conserved across evolution. There is potential to improve our predictions however by incorporating structure. In this paper the authors build a denoising auto-encoder model that incorporates sequence and structure to predict mutation effects. The model is trained to predict the sequence of a protein given its perturbed sequence and structure. The authors demonstrate that this model is able to predict the effects of mutations better than sequence-only models.

As well, the authors curate a set of assays measuring the effect of mutations on thermostability. They demonstrate their model also predicts the effects of these mutations better than previous models and make this benchmark available for the community.

Strengths:

The authors describe a method that makes accurate mutation effect predictions by informing its predictions with structure.

The authors curate a new dataset of assays measuring thermostability. These can be used to validate and interpret mutation effect prediction methods in the future.

Weaknesses:

In the review period, the authors included a previous method, SaProt, that similarly uses protein structure to predict the effects of mutations, in their evaluations. They see that SaProt performs similarly to their method.

ProteinGym is largely made of deep mutational scans, which measure the effect of every mutation on a protein. These new benchmarks contain on average measurements of less than a percent of all possible point mutations of their respective proteins. It is unclear what sorts of protein regions these mutations are more likely to lie in; therefore it is challenging to make conclusions about what a model has necessarily learned based on its score on this benchmark. For example, several assays in this new benchmark seem to be similar to each other, such as four assays on ubiquitin performed in pH 2.25 to pH 3.0.

Comments on revisions:

I think the rounds of review have improved the paper and I've raised my score.

---

## [Author Response]

The following is the authors’ response to the previous reviews.

**Response to Reviewer 1**

Thank you for your recognition of our revised work.

**Response to Reviewer 2**

It would be useful to have a demonstration of where this model outperforms SaProt systematically, and a discussion about what the success of this model teaches us given there is a similar, previously successful model, SaProt.

As two concurrent works, ProtSSN and SaProt employ different methods to incorporate the structure information of proteins. Generally speaking, for two deep learning models that are developed during a close period, it is challenging to conclude that one model is systematically superior to another. Nonetheless, on DTm and DDG (the two low-throughput datasets that we constructed), ProtSSN demonstrates better empirical performance than SaProt.

Moreover, ProtSSN is more efficient in both training and inference compared to SaProt. In terms of training cost, SaProt uses 40 million protein structures for pretraining (requiring 64 A100 GPUs for three months), whereas ProtSSN requires only about 30,000 crystal structures from the CATH database (trained on a single 3090 GPU for two days). Despite SaProt’s significantly higher training cost, its pretrained version does not exhibit superior performance on low-throughput datasets such as DTm, DDG, and Clinvar. Furthermore, the high training cost limits many users from retraining or fine-tuning the model for specific needs or datasets.

Regarding the inference cost, ProtSSN requires only one embedding computation for a wild-type protein, regardless of the number of mutants (n). In contrast, SaProt computes a separate embedding and score for each mutant. For instance, when evaluating the scoring performance on ProteinGym, ProtSSN only needs 217 inferences, while SaProt needs more than 2M inferences. This inference speed is important in practice, such as high-throughput design and screening.

Please remove the reference to previous methods as "few shot". This typically refers to their being trained on experimental data, not their using MSAs. A "few shot" model would be ProteinNPT.

The definition of "few-shot" we used here is following ESM1v [1]. This concept originates from providing a certain number of examples as input to GPT-3 [2]. In the context of protein deep learning models, MSA serves as the wild-type protein examples.

Also, Reviewer 1 uses the concept in the same way.

“Readers should note that methods labelled as "few-shot" in comparisons do not make use of experimental labels, but rather use sequences inferred as homologous; these sequences are also often available even if the protein has never been experimentally tested.”

In the main text, we also included this definition as well as the reference of ESM-1v in lines 457-458.

“We extend the evaluation on ProteinGym v0 to include a comparison of our zero-shot ProtSSN with few-shot learning methods that leverage MSA information of proteins (Meier et al., 2021).”

(1) Meier J, Rao R, Verkuil R, et al. Language models enable zero-shot prediction of the effects of mutations on protein function. Advances in Neural Information Processing Systems, 2021.

(2) Brown T, Mann B, Ryder N, et al. Language models are few-shot learners. Advances in Neural Information Processing Systems, 2020.

Furthermore, I don't think it is fair to state that your method is not comparable to these models -- one can run an MSA just as one can predict a structure. A fairer comparison would be to highlight particular assays for which getting an MSA could be challenging -- Transcription did this by showing that they outperform EVE when MSAs are shallow.

We recognize that there are often differences in the definitions and classifications of various methodologies. Here, we follow the definitions provided by ProteinGym. As the most comprehensive and large scale open benchmark in the community, we believe this classification scheme should be widely accepted. All classifications are available on the official website of ProteinGym (https://proteingym.org/benchmarks), which categorizes methods into PLMs, Structure-based models, and Alignment-based models. For example, GEMME is classified as an alignment-based model, and MSA Transformer is considered a hybrid model combining alignment and PLM features.

We believe that methodologies with different inputs and architectures can lead to inherent unfairness. Also, it is generally believed that models including evolutionary relationships tend to outperform end-to-end models due to the extra information and efforts involved during the training phase. Some empirical evidence and discussions are in the ablation studies of retrieval factors in Tranception [3]. Moreover, the choice of MSA search parameters can introduce uncertainty, which could have positive or negative impacts.

We showcase the impact of MSA depth on model performance with an additional analysis below. Author response image 1 visualizes the Spearman’s correlation between the scores of each model and the number of MSAs on 217 ProteinGym assays, where each point represents one of 217 assays. The summary correlation of each model with respect to all assays are reported in Author response table 1. These results demonstrate no clear correlation between MSA depth and model performance even for MSA-based models.

**Author response image 1. sa3fig1:** Scatter plots of the number of MSA sequences and spearman’s correlation.

**Author response table 1. sa3table1:** Spearmar’s score of the number of MSA sequences and the model’s performance.

Model Name	Model Type	Spearmanr's score
EVE (ensemble)	Alignment-based model	0.239
GEMME	Alignment-based model	0.207
TranceptEVE L	Hybrid - Alignment & PLM	0.237
MSA Transformer (ensemble)	Hybrid - Alignment & PLM	0.262
ESM-IF1	Inverse folding model	0.346
ESM2 (650M)	Protein language model	0.297
ESM-1v (ensemble)	Protein language model	0.372
SaProt (650M)	Hybrid - Structure & PLM	0.260
ProtSSN (ensemble)	Hybrid - Structure & PLM	0.217

(3) Notin P, Dias M, Frazer J, et al. Tranception: protein fitness prediction with autoregressive transformers and inference-time retrieval. International Conference on Machine Learning, 2022.

The authors state that DTm and DDG are conceptually appealing because they come from low-throughput assays with lower experimental noise and are also mutations that are particularly chosen to represent the most interesting regions of the protein. I agree with the conceptual appeal but I don't think these claims have been demonstrated in practice. The cited comparison with Frazer as a particularly noisy source of data I think is particularly unconvincing: ClinVar labels are not only rigorously determined from multiple sources of evidence, Frazer et al demonstrates that these labels are actually more reliable than experiment in some cases. They also state that ProteinGym data doesn't come with environmental conditions, but these can be retrieved from the papers the assays came from. The paper would be strengthened by a demonstration of the conceptual benefit of these new datasets, say a comparison of mutations and signal for a protein that may be in one of these datasets vs ProteinGym.

In the work by Frazer et al. [4], they mentioned that

"However, these technologies do not easily scale to thousands of proteins, especially not to combinations of variants, and depend critically on the availability of assays that are relevant to or at least associated with human disease phenotypes."

It points out that the results of high-throughput experiments are usually based on the design of specific genes (such as BRCA1 and TP53.) and cannot be easily extended to thousands of other genes. At the same time, due to the complexity of the experiment, there may be problems with reproducibility or deviations from clinical relevance.

This statement aligns with our perspective that high-throughput experiments inherently involve a significant amount of noise and error. It is important to clarify that the noise we discuss here arises from the limitations of high-throughput experiments themselves, instead of from the reliability of the data sources, such as systematic errors in experimental measurements. This latter issue is a complex problem common to all wetlab experiments and falls outside the scope of our study.

Under this premise, low-throughput datasets like DTm and DDG can be considered to have less noise than high-throughput datasets, as they have undergone manual curation. As for your suggestion, while valuable, unfortunately, we were unable to identify datasets in DTM and DDG that align with those in ProteinGym after a careful search. Thus, we are unable to conduct this comparative experiment at this stage.

(4) Frazer J, Notin P, Dias M, et al. Disease variant prediction with deep generative models of evolutionary data. Nature, 2021.